# Structure-Based Virtual Screening and De Novo Design of PIM1 Inhibitors with Anticancer Activity from Natural Products

**DOI:** 10.3390/ph14030275

**Published:** 2021-03-18

**Authors:** Hwangseo Park, Jinwon Jeon, Kewon Kim, Soyeon Choi, Sungwoo Hong

**Affiliations:** 1Department of Bioscience and Biotechnology and Institute of Anticancer Medicine Development, Sejong University, 209 Neungdong-ro, Kwangjin-gu, Seoul 05006, Korea; 2Center for Catalytic Hydrocarbon Functionalizations, Institute for Basic Science (IBS), Daejeon 34141, Korea; chistar@kaist.ac.kr (J.J.); kkw105701@kaist.ac.kr (K.K.); sychoi94@kaist.ac.kr (S.C.); 3Department of Chemistry, Korea Advanced Institute of Science and Technology (KAIST), Daejeon 34141, Korea

**Keywords:** virtual screening, de novo design, PIM1, natural products, anticancer activity

## Abstract

Background: the proviral insertion site of Moloney murine leukemia (PIM) 1 kinase has served as a therapeutic target for various human cancers due to the enhancement of cell proliferation and the inhibition of apoptosis. Methods: to identify effective PIM1 kinase inhibitors, structure-based virtual screening of natural products of plant origin and de novo design were carried out using the protein–ligand binding free energy function improved by introducing an adequate dehydration energy term. Results: as a consequence of subsequent enzyme inhibition assays, four classes of PIM1 kinase inhibitors were discovered, with the biochemical potency ranging from low-micromolar to sub-micromolar levels. The results of extensive docking simulations showed that the inhibitory activity stemmed from the formation of multiple hydrogen bonds in combination with hydrophobic interactions in the ATP-binding site. Optimization of the biochemical potency by chemical modifications of the 2-benzylidenebenzofuran-3(2*H*)-one scaffold led to the discovery of several nanomolar inhibitors with antiproliferative activities against human breast cancer cell lines. Conclusions: these new PIM1 kinase inhibitors are anticipated to serve as a new starting point for the development of anticancer medicine.

## 1. Introduction

The proviral insertion site of Moloney murine leukemia (PIM) kinases belong to a serine/threonine kinase family with three isoforms including PIM1, PIM2, and PIM3. Although they play the role of enhancing the proliferation of hematopoietic cells upon stimulation by growth factors and cytokines [1], the enforced expression of PIM kinases was responsible for tumorigenesis in transgenic mouse models [2,3]. In particular, the overexpression of PIM1 has been reported in various human cancers including acute myeloid leukemia and prostate, hepatocellular, colon, and gastric cancers [4,5,6,7,8]. The lack of overt abnormality in PIM1-deficient mice supported the usefulness of PIM1 kinase as a target for the development of new anticancer medicines [9]. PIM1 is also involved in the acquisition of drug resistance by phosphorylating and stabilizing drug efflux transporters such as breast cancer resistance protein and P-glycoprotein [10,11], further motivating the discovery of potent PIM1 inhibitors.

In accordance with the biological and pharmaceutical interest, three-dimensional (3D) structures of PIM1 kinase were reported in complexes with a variety of inhibitors [12,13,14,15,16,17]. The abundance of structural information about interactions between PIM1 and small-molecule ligands is advantageous in designing potent inhibitors that may develop into an anticancer medicine. The three-dimensional structure of PIM1 differs from those of other kinases in that the hinge region at the ATP-binding site adopts an unusual conformation due to the inclusion of proline (Pro123) [12]. Since the backbone amidic nitrogen of Pro123 is incapable of donating a hydrogen bond to the adenine moiety of ATP, conventional ATP mimetic kinase inhibitors are inadequate for tight binding in the ATP-binding site. This is a structural feature useful for the discovery and development of PIM1 inhibitors selective against other kinases [15]. Nonetheless, the discovery of effective PIM1 inhibitors has lagged behind biological and structural studies in that no small-molecule inhibitor has been approved as a new anticancer drug.

Since the identification of staurosporine and bisindolylmaleimides [18], a great deal of scientific endeavors have been devoted to the discovery of structurally diverse PIM1 inhibitors. They include imidazo[1,2-b]pyridazine [19], benzoisoxazole [20], thiazolidinone [21], cinnamic acid [22], benzofuranone [23,24], picolinamide [25], 4-azapodophyllotoxin [26], quinone [27], and azaindole [28] moieties as the molecular core. Complementary to the experimental discovery, computational studies for PIM1–inhibitor complexes have also been actively pursued by means of quantitative structure–activity relationship [29,30], docking simulations [31], and quantum chemical calculations [32] to seek the rationale for inhibitory activities. Despite the high biochemical potency, most PIM1 inhibitors failed to develop into a new anticancer drug due in a large part to poor pharmacological properties. In this regard, it would be necessary to find a promising PIM1 inhibitor scaffold from natural resources including plants. Natural products have indeed provided the molecular cores to acquire desirable pharmacological properties to the extent that they take a large portion of drugs in the market [33].

This work was undertaken to find a new class of PIM1 inhibitors through virtual screening of natural products with molecular docking and de novo design from the initial hit molecules. Virtual screening and de novo design with docking simulations have often been unsuccessful due to the roughness of the protein–ligand binding energy function to score and rank the putative inhibitors [34,35]. In particular, the underestimation of ligand dehydration in protein–ligand binding has been invoked to elucidate the poor correlation between computational estimations of biochemical potency and the corresponding experimental measurements [36]. Accordingly, the potential-based scoring function including a sophisticated molecular hydration energy term showed the outperformance in virtual screening for various target proteins by alleviating overestimation of the biochemical potency of a ligand with hydrophilic moieties [37,38]. It is further exemplified in this work that even low-nanomolar PIM1 inhibitors with anticancer activity can be identified from natural products using the modified scoring function.

## 2. Results and Discussion

Among a total of 42,613 natural products in the chemical database, 31,874 molecules that satisfied the physicochemical property criteria for a potential drug candidate [39] were screened through docking simulations in the ATP-binding site of PIM1 kinase, as illustrated in Figure 1. Using the modified binding free energy function involving an accurate molecular dehydration term, 100 top-scoring compounds were selected as the virtual hits, all of which were commercially available from a compound vendor (InterBioScreen Ltd., https://www.ibscreen.com, accessed on 15 October 2019). All these putative inhibitors were tested for the presence of the inhibitory activity against PIM1 kinase with in vitro radiometric ([γ-^33^P]-ATP) kinase assays (Reaction Biology Corp., Malvern, PA, USA). As a consequence of the virtual and high-throughput screening processes, four natural products were identified as new PIM1 inhibitors. Each natural-product inhibitor revealed a good biochemical potency against PIM1, with percent of control (POC) values lower than 10 at 100 μM, and were determined to measure the IC_50_ values.

The chemical structures and biochemical potencies of the four natural-product PIM1 inhibitors are summarized in Figure 2 and Table 1, respectively. It is worth noting that all four inhibitors (**1–4**) have origins from plant sources. For example, **1** is a constituent of *Scutellaria ramosissima* (Lamiaceae), with the flavonoid scaffold as the molecular core, while **2** is isolated from soja and retama (*Lygos* spp.). Inhibitors **3** and **4** also originate from plant organisms, *Xanthoceras sorbifolia* and *Sophora japonica*, respectively. It is a common structural feature of **1–4** that several hydrogen bonding moieties reside on the fused ring including a phenyl group in the vicinity of the other phenyl ring. This implies that all four PIM1 inhibitors would be accommodated in the ATP-binding site through the multiple hydrogen bonds along with the hydrophobic interactions. In a pharmacological point of view, **1** and **2** seem to be more druggable than **3** and **4** owing to the possession of less than five hydrogen-bond donors.

As listed in Table 1, **1–3** exhibit good inhibitory activity against PIM1 at low-micromolar to sub-micromolar levels while **4** has relatively moderate biochemical potency. All these four natural-product inhibitors deserve consideration for further development to optimize their anticancer activities by chemical modifications because they were screened virtually for possessing physicochemical properties as a drug candidate. In particular, **1** and **2** seem to be promising in the context that they have relatively high biochemical potency in spite of low molecular weights at 314.3 and 268.3 amu, respectively. Although the inhibitory activity of **2** is a little lower than that of **1**, the former is anticipated to serve as the better molecular scaffold than the latter for potency optimization by derivatizations due to the possession of much more substitution points.

To gain structural insight relevant to the inhibitory activities of the newly identified natural-product inhibitors, their binding modes in the ATP-binding site of PIM1 were examined in the comparative fashion. Figure 3 shows the most stable binding configurations of **1–4** derived with docking simulations using the 3D structure of PIM1 in complex with a potent inhibitor (Protein Data Bank (PDB) code: 5DWR) [25]. Although some differences are observed in the detailed interaction patterns, **1–4** appear to be accommodated in a similar way in the ATP-binding site of PIM1. For example, the polar moieties on the fused ring are directed toward the backbone groups of the hinge region with a unique LERPXPX motif that involves the gatekeeper residue (Leu120). On the other hand, the hydrophobic aromatic rings reside in proximity to the glycine-rich phosphate-binding loop (Gly loop). The necessity of the interactions with the hinge region and Gly-loop (residues 46–54) for tight binding to PIM1 was also implicated in the X-ray crystallographic studies of PIM1 in complex with potent inhibitors [40,41,42]. Inhibitors **1–4** also appear to interact with the activation loop at the top of the C-terminal domain (residues 129–305), which includes the conserved DFG (Asp186-Phe187-Gly188) motif.

As a check on the plausibility of the binding modes that differ from those in Figure 3, the additional docking simulations of **1–4** were conducted extensively with respect to PIM1 using the 3D grid maps constructed to encompass the entire kinase domain. The results for clustering analyses of a total of 100 docking runs for all four protein–ligand complexes showed that the binding modes in Figure 3 represented the lowest binding free energy and, simultaneously, the most probable cluster, with more than 57% of the total population. Furthermore, no peripheral binding pocket was identified during the entire course of docking simulations in which **1–4** could be accommodated with a negative value of binding free energy. These results support the reliability of the aforementioned binding modes of the four natural-product inhibitors of PIM1.

To address the structural relevance to the biochemical potencies ranging from low-micromolar to sub-micromolar levels, the calculated binding modes of the newly found natural-product inhibitors in the ATP-binding site of PIM1 were analyzed in detail. The lowest-energy binding poses of **1** and **2** with respect to PIM1 are compared in Figure 4. Both in the PIM1–**1** and PIM1–**2** complexes, the phenolic –OH moieties of the inhibitors appear to play the role of an anchor for binding through the formation of the two hydrogen bonds in the ATP-binding site. For example, the two vicinal –OH groups of **1** donate a hydrogen bond to the backbone aminocarbonyl oxygen of Glu121 (Figure 4a), which is located in the middle of the hinge region between the N- and C-terminal domains. The appearance of these bifurcated hydrogen bonds is consistent with earlier X-ray crystallographic studies of PIM1–inhibitor complexes in which the formation of a strong hydrogen bond with Glu121 in the hinge region is necessary for tight binding in the ATP-binding site [43,44]. Inhibitor **1** may be classified as an ATP-mimetic inhibitor on the grounds that it forms hydrogen bonds only with Glu121 instead of the amino acid residues in the activation loop and the catalytic loop.

The binding mode of **2** differs from that of **1** in that the terminal phenolic moiety plays the role of a hydrogen-bond donor with respect to the side-chain carboxylate ion of Glu89 on the catalytic loop (Figure 4b) instead of Glu121 in the hinge region. This hydrogen bond seems to be stabilized by the other hydrogen bond established between the phenolic –OH moiety of **2** and the backbone amidic nitrogen of Phe187, a component of the conserved DFG (Asp186-Phe187-Gly188) motif on the activation loop. This interaction has the effect of polarizing the phenolic O–H bond by partial protonation, which culminates in the strengthening of the subsequent hydrogen bond with Glu89. Inhibitor **2** can thus be classified as a non-ATP mimetic ATP-competitive inhibitor on the grounds that it forms hydrogen bonds with the amino acid residues in the activation loop and the catalytic loop instead of those in the hinge region. Although the calculated PIM1–**1** and PIM1–**2** complexes have the same number of hydrogen bonds, those in the former would be stronger than those in the latter because the phenolic –OH moiety tends to act as a hydrogen bond donor rather than as an acceptor due to poor basicity. Hence, strengthening of the hydrogen bonds may be invoked to explain the higher inhibitory activity of **1** than **2**.

In contrast to a large difference in the hydrogen-bonding patterns between PIM1–**1** and PIM1–**2** complexes, the two natural-product inhibitors appear to form hydrophobic interactions with almost the same amino-acid residues in the ATP-binding pocket. As can be seen in Figure 4, for example, nonpolar moieties of both **1** and **2** form the van der Waals contacts with the hydrophobic side chains of Leu44, Val52, Ala65, Leu93, Ile104, Leu120, Val126, Leu174, Ile185, and Phe187. Notably, the phenyl rings involving the –OH moieties are accommodated by the hydrophobic residues of the Gly loop (Leu44, Val52, and Ala65) in the PIM1–**1** complex, as compared to those at the interface of N- and C-terminal domains (Ile104, Leu120, and Ile185) in the PIM1–**2** complex. These hydrophobic interactions are likely to play the role of protecting the neighboring hydrogen bonds from disruptive solvent molecules. Some synergistic effects may thus be produced in binding affinity due to the formation of the two hydrogen bonds in the vicinity of the hydrophobic contacts. In this regard, it has been widely adopted in the potency optimization of drug candidates to strengthen the hydrogen bonds along with the proximal hydrophobic interactions with a target protein [45,46]. Taken together, the respective sub-micromolar and low-micromolar inhibitory activities of **1** and **2** against PIM1 can be attributed to the multiple hydrogen bonds and hydrophobic interactions maintained cooperatively in the ATP-binding site.

Despite the lower inhibitory activity of **2** than **1**, the former is preferred over the latter as the molecular core for the potency optimization in terms of synthetic availability for preparing the chemical derivatives. With respect to enhancing the biochemical potency using chemical modifications, we note that **2** cannot form a hydrogen bond with the amino acid residues in the hinge region (Figure 4b), which may be invoked to explain the lower inhibitory activity than **1**. Hence, the introduction of a hydrogen-bonding group at a proper substitution position of **2** enhances the biochemical potency by inducing interactions with the hinge region. It is also noteworthy that the anisole moiety of **2** is not fully accommodated in the hydrophobic pocket comprising Leu44, Val52, Pro123, Val126, and Leu174 (Figure 4b). A more potent inhibitor than **2** is therefore anticipated if a hydrophobic group of the increased van der Waals volume is substituted for the terminal anisole moiety. The substitution of a hydrophobic moiety has an advantage over a hydrogen-bonding group because the former has little impact on the hydration cost while the latter tends to decrease the binding affinity for the target protein due to the increased stabilization in water. Using **2** as the molecular scaffold, we carried out the structure–activity relationship study via de novo design and chemical syntheses of the designed molecules.

A variety of putative PIM1 inhibitors were generated in the de novo design with 2-methylenebenzofuran-3(2*H*)-one (**C**) as the molecular core. In this step, we attempted to find druggable PIM1 inhibitors through simultaneous optimization of the inhibitory activity against PIM1 and drug-like properties. Only the highly scored derivatives of **C** were then prepared by chemical synthesis, as illustrated in Scheme 1. The aurone derivatives **C** were readily obtained by condensation of benzofuran-3(2*H*)-one **A** with various aldehydes **B** under basic or acidic reaction conditions.

Table 2 lists the structures and IC_50_ values of various 2-methylenebenzofuran-3(2*H*)-one derivatives that were selected for chemical synthesis among the virtual hits of de novo design. The substitution points included C4 (R1) and C6 (R2) positions of the benzofuran-3(2*H*)-one ring as well as the β carbon (R3) with respect to the carbonyl group. It is interesting to note that three derivatives (**5–7**) were previously reported as the inducers of NAD(P)H:quinone oxidoreductase 1 (NQO1) [47]. Overall, eight of the twelve synthesized derivatives exhibit high biochemical potency at the sub-micromolar level against PIM1. This result exemplifies the accuracy of the modified binding free energy function with which the new PIM1 inhibitors were designed so as to maximize the binding affinity and simultaneously to minimize the dehydration cost for binding in the ATP-binding site.

In Table 2, it is worth noting that the majority of the newly synthesized inhibitors include the polar aromatic groups at the R3 position connected to the benzofuran-3(2*H*)-one ring with the ethylene linker. Because the anisole group of **2** resides at the end of the hinge region (Figure 4b), the substituted aromatic moieties seem to interact with the amino acid residues at the entrance of the ATP-binding site. The movement of the terminal methoxy moiety of **2** from the *para* to *meta* position in **5** leads to an almost ten-fold increase in the biochemical potency, while the additional substitution of a hydroxyl moiety at the R1 position in **6** has little effect on the inhibitory activity. Although the substitution of a chlorine atom in **7** for the methoxy group in **6** leads to an increase in the IC_50_ value, the sub-micromolar inhibitory activities are restored in **8–10** due to the introduction of a chlorine atom as the second substituent in the vicinity of the existing methoxy and chlorine substituent. As can be inferred from the similar IC_50_ values between **5** and **6** as well as between **8** and **9**, the biochemical potency remains almost unchanged with the substitution of a hydroxyl group at the R1 position.

It is interesting to note that the inhibitory activity decreases by a factor of 20 due to replacement of the terminal phenyl ring in **5** with pyridine in **11**. Similar results are also obtained by introducing polar heterocyclic groups including pyridine-2(1*H*)-one (**12**) and quinoline-2(1*H*)-one (**13**) at the R3 position. Apparently, such a reduction in the inhibitory activity in **11–13** can be attributed to the greater increase in the hydration cost than the strengthening of binding to PIM1. This indicates that the accurate calculation of dehydration energy is necessary for designing potent PIM1 inhibitors as well as the binding force in the ATP-binding site.

The inhibitory activity surges to the nanomolar level upon substitution of aromatic heterocycles such as 4-(1*H*-pyrazol-5-yl)pyridine in **14**, 6-chloro-1*H*-pyrrolo[2,3-*b*]pyridine in **15**, and 2-chloro-7*H*-pyrrolo[2,3-*d*]pyrimidine in **16** at the R3 position. As can be inferred from the decrease in IC_50_ values going from **14** to **15** and **16**, the fused aromatic rings is preferred to bicyclic substituents. The achievement of nanomolar inhibitory activity is remarkable because **14–16** contain a more polar group than **2** at both the R1 and R3 positions, which would have the effect of reducing biochemical potency due to the increased dehydration cost. In this regard, it is most likely that one can augment the inhibitory activity with structural modifications in such a way to make the interactions with PIM1 strong enough to surmount the increased dehydration cost for binding in the ATP-binding site.

Among a variety of the synthesized derivatives of **2**, the most potent PIM1 inhibitor **16** was obtained with substitutions of –OH and 2-chloro-7*H*-pyrrolo[2,3-*d*]pyrimidine moiety at the R1 and R3 positions, respectively, with the associated IC_50_ value of 7 nM. Figure 5 shows the lowest-energy binding mode of **16** derived from docking simulations in the ATP-binding site. It is a common structural feature in the calculated PIM1–**2** and PIM1–**16** complexes that one phenolic –OH moiety forms two hydrogen bonds with Phe187 and Glu89 along with the hydrophobic interactions with nonpolar residues in the ATP-binding site. Consistent with the low-nanomolar inhibitory activity, some additional potency-enhancing interactions are observed in the PIM1–**16** complex. For instance, the second phenolic –OH moiety introduced at the R1 position donates a hydrogen bond to the sidechain carboxylate group of Asp186 on the conserved DFG (Asp186-Phe187-Gly188) motif. Apparently, this additional hydrogen bond enhances the biochemical potency of **16** by binding more tightly to PIM1. The inhibitory action of **16** is similar to those of CX-4945 and Ro-3306 in the involvement of hydrogen-bond interactions with Asp186 [17]. Inhibitor **16** appears to be further stabilized in the ATP-binding site through the fourth hydrogen bond in the hinge region. More specifically, the backbone amidic nitrogen of Pro123 serves as the hydrogen-bond acceptor with respect to the 7*H*-pyrrolo[2,3-*d*]pyrimidine ring of **16** in the calculated PIM1–**16** complex (Figure 5). Besides the increase in the number of hydrogen bonds, van der Waals interactions also become stronger going from the PIM1–**2** to PIM1**–6** complexes due to the enlargement of the molecular volume at the R3 position. Judging from the docking simulation results, the low-nanomolar inhibitory activity of **16** can be attributed to the consolidation of both hydrogen-bond and van der Waals interactions. Since the substitution of 2-chloro-7*H*-pyrrolo[2,3-*d*]pyrimidine moiety in **16** causes increased stabilization in aqueous solution, the strengthening of the interactions in the ATP-binding site seems to be sufficient to overcome the increased dehydration cost for binding to PIM1.

To assess the anticancer activities of these new PIM1 inhibitors, we carried out cell-based assays for **5**, **8**, **14**, **15**, and **16** using T47D human cancer cell lines, which represented breast cancer caused by PIM1 [48]. The natural product wortmannin served as a positive control in this cellular study. Cell viabilities were measured at varying inhibitor concentrations with a 3-(4,5-dimethylthiazol-2-yl)-2,5-diphenyltetrazolium bromide (MTT) assay. As shown in Table 3, all five inhibitors reveal significant biochemical potency against the proliferation of T47D cell lines with associated IC_50_ values ranging from 3 to 15 μM. The presence of antiproliferative activity indicates that the newly found PIM1 inhibitors may serve as a starting point for the development of new anticancer medicines. However, it should be noted that the inhibitory activities of **14–16** decrease from the nanomolar level in enzyme inhibition assays (Table 2) to the micromolar level in cell proliferation assays. The poor cell permeability can be invoked to explain the reduced biochemical potency in cell-based assays [49,50]. In order for the new PIM1 inhibitors to become a good lead compound for the development of anticancer medicines, improving cellular permeability using further structural modifications is required.

Although several promising PIM1 inhibitors were discovered from natural products, the inhibitory activities of some compounds in Table 1 and Table 2 remain moderate despite modification of the protein–ligand binding energy function for virtual screening and de novo design. This stems from the imperfection of the scoring function, which can be ascribed in a large part to incomplete optimization of the weighting factors for individual energy terms. This is actually inevitable because the number of known PIM1–inhibitor complexes is insufficient to constitute a training set appropriate for parameterizations. The protein–ligand binding energy function is anticipated to become even more accurate once the weighting factors are reoptimized after supplementing the training set with a variety of the new PIM1–inhibitor complexes. We plan to design and synthesize the more potent PIM1 inhibitors with increased antiproliferative activity compared to those presented in this work using the further improved scoring function.

## 3. Materials and Methods

### 3.1. Preparation of the Atomic Coordinates of PIM1 and the Natural Product Library

Three-dimensional atomic coordinates of the receptor model for structure-based virtual screening and de novo design were prepared from the X-ray crystal structure of PIM1 in complex with a potent and selective small-molecule inhibitor (PDB code: 5DWR) [26]. After removal of the crystallographic solvent molecules, the hydrogen atoms were added to each protein atom to complete the all-atom model for PIM1. To this end, the protonation states of all ionizable amino-acid residues (Asp, Glu, His, and Lys) were assigned under consideration of the intramolecular hydrogen-bonding patterns in the original X-ray crystal structure. For example, the sidechains of Asp and Glu residues were kept protonated if either of the carboxylate oxygen atoms pointed toward a hydrogen-bond accepting group at a distance within 3.5 Å. Similarly, the sidechain butylamine group of a lysine residue was assumed to be protonated unless it played the role of hydrogen-bond acceptor in the original X-ray crystal structure. The same procedure was applied to assign the protonation state of the imidazole moiety of all histidine residues. After the addition of all the hydrogen atoms to produce the 3D receptor model, a total of 2000 cycles of energy minimization were carried out to remove all bad steric contacts caused by the increase in atomic population.

Prior to conducting docking simulations for virtual screening, a molecular library was prepared from a total of 42,613 natural products in the chemical database distributed by a compound vendor (InterBioScreen Ltd., https://www.ibscreen.com). All these natural products were filtrated according to the “Rule of Five” using the ISIS/BASE program to screen the molecules with desirable physicochemical properties as a potential drug candidate [39]. To avoid structural redundancy in the selected molecules, similar natural products with an associated Tanimoto coefficient larger than 0.8 were clustered to a single representative one. A total of 31,874 natural products selected with the two-step filtrations were then processed with the CORINA program to generate the 3D atomic coordinates [51], which was followed by the atomic charge calculations using the Gasteiger–Marsilli method [52].

### 3.2. Virtual Screening to Identify the PIM1 Inhibitors of Natural Origin

To identify new PIM1 inhibitors from the natural product library, virtual screening was conducted using the AutoDock program [53] to collect the candidates through the docking simulations in the ATP-binding site. Despite a significant contribution to the protein–ligand association, it was difficult to reflect the ligand hydration effects explicitly in docking simulations because the scoring function of the original AutoDock program contained a crude dehydration energy term involving only six atom types for varying solute molecules. Therefore, the modified version of AutoDock program was used in this work because the outperformance of its scoring function was demonstrated in various target proteins [37,38]. This modified scoring function ($\Delta G_{bind}^{aq}$) differs from the original one in the inclusion of a sophisticated dehydration free energy term, which can be expressed in the following mathematical form.
(1)$$\begin{array}{ll} \Delta G_{bind}^{aq} & =W_{vdW}\sum_{i=1}\sum_{j>i}\left(\frac{A_{ij}}{r_{ij}^{12}}-\frac{B_{ij}}{r_{ij}^{6}}\right)+\;W_{hbond}\sum_{i=1}\sum_{j>i}E(t)\left(\frac{C_{ij}}{r_{ij}^{12}}-\frac{D_{ij}}{r_{ij}^{10}}\right)\;\\ & +\;W_{elec}\sum_{i=1}\sum_{j>i}\frac{q_{i}q_{j}}{\varepsilon (r_{ij})r_{ij}}\;+\;W_{tor}N_{tor}\;+\;W_{sol}\sum_{i=1}S_{i}\left(Occ_{i}^{\max}-\sum_{j\neq i}V_{j}e^{-\frac{r_{ij}^{2}}{2\sigma^{2}}}\right) \end{array}$$

The coefficients *W_vdW_*, *W_hbond_*, and *W_elec_* in Equation (1) refer to the weighting factors of van der Waals, hydrogen bond, and electrostatic interactions, respectively, between PIM1 and ligand atoms. *W_tor_* is the coefficient of the torsional motions of a putative inhibitor, while *W_sol_* is associated with the ligand dehydration cost for binding to PIM1. The variable *r_ij_* denotes the interatomic separation, and the *A_ij_*, *B_ij_*, *C_ij_*, and *D_ij_* parameters determine the well depth and the equilibrium distance for a given potential energy function. AMBER force field parameters were used to calculate the van der Waals interaction energies between PIM1 and a putative inhibitor. The hydrogen-bond energy term contains an additional weighting factor (*E*(*t*)) to describe the angle-dependent directionality. In calculating the intermolecular electrostatic interaction energies between PIM1 and a putative inhibitor, the atomic charges obtained using the Gasteiger–Marsilli method [52] were used along with the distance-dependent sigmoidal function as the dielectric constant for the long-range charge screening effects [54]. The *N_tor_* parameter in the torsional term is equal to the number of rotatable bonds, which is adopted to estimate the entropic penalty for a putative inhibitor to bind to PIM1.

The sum of the first four terms in Equation (1) correspond to the protein–ligand binding energy in the gas phase, whereas the final term is the negative of the ligand hydration energy. In this dehydration term, the *S_i_*, *V_i_*, and *O_i_^max^* parameters stand for the atomic hydration energy per unit volume, the atomic volume in molecules, and the maximum atomic occupancy, respectively [55]. To estimate the hydration free energies of each putative PIM1 inhibitor, all atomic parameters were derived with the extended solvent-contact model [56]. Substitution of this new hydration free energy term in the scoring function has the effect of enhancing the accuracy of virtual screening by preventing overestimation of the binding affinity of a candidate inhibitor with many polar groups.

Using the modified AutoDock scoring function in Equation (1), docking simulations were carried out in the ATP-binding site PIM1 to score and rank the natural products according to the calculated binding affinities. Only the natural products included in the 100 top-ranked virtual hits were selected for subsequent biochemical evaluations.

### 3.3. De Novo Design

To improve the biochemical potency of the new PIM1 inhibitors, the initial hit compounds identified from virtual screening were structurally evolved to maximize the interactions in the ATP-binding site. To this end, structure-based de novo designs were performed in a stepwise fashion. First, a variety of derivatives of a hit compound were generated with the LigBuilder program [57] using the structure of PIM1 in complex with the hit compound as the input structure. This procedure proceeded with the genetic algorithm to improve the structure of the molecular core by substituting various chemical moieties at specified positions. The number of substitution positions was limited to three in this work to reduce the computational cost. The empirical scoring function consisting of electrostatic, van der Waals, hydrogen bond, and entropic terms was used to select the derivatives that were estimated to bind more tightly to PIM1 than the initial hit. The bioavailability rules were also applied to collect only derivatives with druggable physicochemical properties.

The second step of de novo design was performed in the same way as the precedent virtual screening in the context that all the derivatives generated in the first step were further screened using the modified scoring function in Equation (1). The derivatives with a binding free energy at 5 kcal/mol lower than the initial hit were then inspected for the availability of chemical synthesis. Finally, twelve derivatives of the initial hit were synthesized and evaluated with enzyme inhibition assays to identify the new potent PIM1 inhibitors.

### 3.4. Chemical Synthesis

#### 3.4.1. General Methods

Unless stated otherwise, all chemical reactions were performed in flame-dried glassware. Analytical thin layer chromatography (TLC) was performed on precoated silica gel 60 F^254^ plates, and visualization on TLC was achieved by UV light (254 and 365 nm). A high-performance liquid chromatography (HPLC) instrument was performed using a Shimadzu HPLC system (pump: LC-20AP, module: CBM-20A, UV/Vis detector: SPD-20A, fraction collector: FRC-10A, column: Shim-pack GIS C18 column, 10 × 250 mm, 5 μm particle size). ^1^H NMR was recorded on Brucker Avance 400 MHz or Agilent Technologies DD2 600 MHz, and chemical shifts were quoted in parts per million (ppm), referenced to 2.50 ppm for DMSO-*d*_6_. The following abbreviations were used to describe peak splitting patterns when appropriate: br = broad, s = singlet, d = doublet, t = triplet, q = quartet, m = multiplet, dd = doublet of doublet, td = triplet of doublet, and ddd = doublet of doublet of doublet. Coupling constants, *J*, were reported in hertz unit (Hz). ^13^C NMR was recorded on Brucker Avance 100 MHz or Agilent Technologies DD2 150 MHz and was fully decoupled by broad band proton decoupling. Chemical shifts of ^13^C NMR were reported in ppm, referenced to the centerline of a septet at 39.52 ppm of DMSO-*d*_6_. High-resolution mass spectra were obtained by the electrospray ionization (ESI) method from KAIST Research Analysis Center (Daejeon, Korea), which involved the mass measurement with a Quadrupole-TOF MS system.

#### 3.4.2. General Procedure I (GPI) (Compound 5–14)

Aldehyde (1.2 equiv.) and aq. 4N KOH (20 equiv.) were added to a solution of benzofuranone derivative (1 equiv.) in MeOH (0.1 M). The reaction was stirred at 110 °C for 30 min under microwave irradiation. The reaction mixture was monitored by TLC. After the disappearance of the starting material, the reaction mixture was concentrated under reduced pressure. The residue was diluted with distilled water and acidified with 1N HCl until pH reached 1–2 in an ice bath. The precipitated solid was filtered while washing with distilled water and CH_2_Cl_2_ to give the corresponding desired product as a yellow solid. See the Appendix A for spectral copies.

#### 3.4.3. General Procedure II (GPII) (Compound 15 and 16)

Aldehyde (1.0 equiv.) and aq. 12N HCl (0.1 equiv.) were added to a solution of benzofuranone derivative (1 equiv.) in EtOH (0.25 M). The reaction was stirred at 80 °C for 3 h. The reaction mixture was monitored by TLC. After the disappearance of the starting material, the reaction mixture was concentrated under reduced pressure. The residue was diluted with distilled water and was filtered while washing with distilled water and CH_2_Cl_2_ to give the corresponding desired product as a yellow solid. Compound **16** was further purified by prep-HPLC (MeOH/H_2_O).

(*Z*)-6-Hydroxy-2-(3-methoxybenzylidene)benzofuran-3(2*H*)-one (**5**). Compound **5** (62.8 mg, 70%) was obtained according to GPI using 3-methoxybenzaldehyde and 6-hydroxybenzofuran-3(2*H*)-one. Yellow solid. ^1^H NMR (600 MHz, DMSO-*d*_6_) δ 11.23 (s, 1H), 7.63 (d, *J* = 8.4 Hz, 1H), 7.55 (d, *J* = 7.5 Hz, 1H), 7.50 (s, 1H), 7.41 (t, *J* = 7.9 Hz, 1H), 7.02 (d, *J* = 8.1 Hz, 1H), 6.81 (s, 1H), 6.76 (s, 1H), 6.73 (d, *J* = 8.4 Hz, 1H), 3.81 (s, 3H). ^13^C NMR (100 MHz, DMSO-*d*_6_) δ 181.5, 168.0, 166.7, 159.4, 147.5, 133.3, 130.0, 126.0, 123.4, 116.4, 115.3, 113.1, 112.7, 110.3, 98.7, 55.2. LCMS (ESI^+^) *m/z* calcd. For [C_16_H_12_NaO_4_]^+^: 291.0633, found: 291.0641.

(*Z*)-4,6-Dihydroxy-2-(3-methoxybenzylidene)benzofuran-3(2*H*)-one (**6**). Compound **6** (36.7 mg, 21%) was obtained according to GPI using 3-methoxybenzaldehyde and 4,6-dihydroxybenzofuran-3(2*H*)-one. Yellow solid. ^1^H NMR (400 MHz, DMSO-*d*_6_) δ 10.99 (s, 2H), 7.49 (d, *J* = 7.7 Hz, 1H), 7.44 (s, 1H), 7.38 (t, *J* = 7.9 Hz, 1H), 6.98 (dd, *J* = 8.1, 2.1 Hz, 1H), 6.57 (s, 1H), 6.19 (s, 1H), 6.04 (s, 1H), 3.80 (s, 3H). ^13^C NMR (125 MHz, DMSO-*d*_6_) δ 179.0, 167.9, 167.8, 159.4, 148.0, 133.7, 129.9, 123.0, 116.0, 114.8, 107.9, 102.6, 98.0, 90.5, 55.1. LCMS (ESI^+^) *m/z* calcd. For [C_16_H_12_NaO_5_]^+^: 307.0582, found: 307.0593.

(*Z*)-2-(3-Chlorobenzylidene)-4,6-dihydroxybenzofuran-3(2*H*)-one (**7**). Compound **7** (45.5 mg, 87%) was obtained according to GPI using 3-chlorobenzaldehyde and 4,6-dihydroxybenzofuran-3(2*H*)-one. Yellow solid. ^1^H NMR (400 MHz, DMSO-*d*_6_) δ 11.04 (s, 1H), 10.97 (s, 1H), 7.92 (s, 1H), 7.87 (d, *J* = 7.5 Hz, 1H), 7.49 (t, *J* = 7.8 Hz, 1H), 7.44 (d, *J* = 7.8 Hz, 1H), 6.62 (s, 1H), 6.24 (s, 1H), 6.10 (s, 1H). ^13^C NMR (100 MHz, DMSO-*d*_6_) δ 178.9, 167.9, 167.7, 158.6, 148.4, 134.6, 133.5, 130.7, 129.9, 129.1, 128.8, 106.4, 102.3, 97.9, 90.8. LCMS (ESI^+^) *m/z* calcd. For [C_15_H_9_ClNaO_4_]^+^: 311.0087, found: 311.0087.

(*Z*)-2-(4-Chloro-3-methoxybenzylidene)-6-hydroxybenzofuran-3(2*H*)-one (**8**). Compound **8** (54.4 mg, 90%) was obtained according to GPI using 4-chloro-3-methoxybenzaldehyde and 6-hydroxybenzofuran-3(2*H*)-one. Yellow solid. ^1^H NMR (400 MHz, DMSO-*d*_6_) δ 11.29 (s, 1H), 7.72–7.56 (m, 3H), 7.52 (d, *J* = 8.3 Hz, 1H), 6.81 (d, *J* = 1.7 Hz, 1H), 6.79 (s, 1H), 6.73 (dd, *J* = 8.5, 1.7 Hz, 1H), 3.92 (s, 3H). ^13^C NMR (100 MHz, DMSO-*d*_6_) δ 181.3, 167.9, 166.8, 154.6, 147.6, 132.4, 130.3, 126.0, 123.9, 122.4, 115.0, 113.2, 112.6, 109.4, 98.8, 56.1. LCMS (ESI^+^) *m/z* calcd. For [C_16_H_11_ClNaO_4_]^+^: 325.0244, found: 325.0241.

(*Z*)-2-(4-Chloro-3-methoxybenzylidene)-4,6-dihydroxybenzofuran-3(2*H*)-one (**9**). Compound **9** (53.7 mg, 93%) was obtained according to GPI using 4-chloro-3-methoxybenzaldehyde and 4,6-dihydroxybenzofuran-3(2*H*)-one. Yellow solid. ^1^H NMR (400 MHz, DMSO-*d*_6_) δ 11.01 (s, 1H), 10.94 (s, 1H), 7.62 (d, *J* = 1.5 Hz, 1H), 7.54 (dd, *J* = 8.3, 1.5 Hz, 1H), 7.50 (d, *J* = 8.3 Hz, 1H), 6.63 (s, 1H), 6.24 (d, *J* = 1.7 Hz, 1H), 6.09 (d, *J* = 1.7 Hz, 1H), 3.91 (s, 3H). ^13^C NMR (100 MHz, DMSO-*d*_6_) δ 178.9, 167.8, 167.6, 158.5, 154.6, 148.0, 132.7, 130.2, 123.5, 121.8, 114.7, 107.3, 102.4, 97.9, 90.8, 56.1. LCMS (ESI^+^) *m/z* calcd. For [C_16_H_12_ClO_5_]^+^: 319.0373, found: 319.0373.

(*Z*)-2-(3,4-Dichlorobenzylidene)-4,6-dihydroxybenzofuran-3(2*H*)-one (**10**). Compound **10** (15.0 mg, 24%) was obtained according to GPI using 3,4-dichlorobenzaldehyde and 4,6-dihydroxybenzofuran-3(2*H*)-one. Yellow solid. ^1^H NMR (400 MHz, DMSO-*d*_6_) δ 11.04 (s, 2H), 8.10 (d, *J* = 1.9 Hz, 1H), 7.89 (dd, *J* = 8.5, 2.0 Hz, 1H), 7.72 (d, *J* = 8.4 Hz, 1H), 6.63 (s, 1H), 6.24 (d, *J* = 1.7 Hz, 1H), 6.09 (d, *J* = 1.7 Hz, 1H). ^13^C NMR (100 MHz, DMSO-*d*_6_) δ 178.7, 167.8, 167.8, 158.6, 148.7, 133.3, 131.9, 131.6, 131.3, 131.1, 130.3, 105.4, 102.3, 98.0, 90.8. LCMS (ESI^+^) *m/z* calcd. For [C_15_H_9_Cl_2_O_4_]^+^: 322.9878, found: 322.9887.

(*Z*)-6-Hydroxy-2-((6-methoxypyridin-3-yl)methylene)benzofuran-3(2*H*)-one (**11**). Compound **11** (83.0 mg, 93%) was obtained according to GPI using 6-methoxynicotinaldehyde and 6-hydroxybenzofuran-3(2*H*)-one. Yellow solid. ^1^H NMR (400 MHz, DMSO-*d*_6_) δ 11.21 (s, 1H), 8.66 (s, 1H), 8.40–8.18 (m, 1H), 7.60 (d, *J* = 8.4 Hz, 1H), 6.93 (d, *J* = 8.7 Hz, 1H), 6.78 (s, 2H), 6.70 (d, *J* = 8.4 Hz, 1H), 3.90 (s, 3H). ^13^C NMR (100 MHz, DMSO-*d*_6_) δ 181.1, 167.8, 166.5, 163.7, 150.4, 147.1, 140.3, 125.9, 122.1, 113.1, 112.8, 111.1, 107.4, 98.7, 53.5. LCMS (ESI^+^) *m/z* calcd. For [C_15_H_11_NNaO_4_]^+^: 292.0586, found: 292.0588.

(*Z*)-3-((4,6-Dihydroxy-3-oxobenzofuran-2(3*H*)-ylidene)methyl)pyridin-2(1*H*)-one (**12**). Compound **12** (114 mg, 70%) was obtained according to GPI using 2-oxo-1,2-dihydropyridine-3-carbaldehyde and 4,6-dihydroxybenzofuran-3(2*H*)-one. Yellow solid. ^1^H NMR (400 MHz, DMSO-*d*_6_) δ 12.00 (s, 1H), 10.94 (s, 1H), 10.89 (s, 1H), 8.26 (d, *J* = 7.1 Hz, 1H), 7.47 (d, *J* = 5.5 Hz, 1H), 6.81 (s, 1H), 6.37 (t, *J* = 6.8 Hz, 1H), 6.20 (s, 1H), 6.07 (s, 1H). ^13^C NMR (100 MHz, DMSO-*d*_6_) δ 178.7, 167.5, 167.3, 161.4, 158.4, 148.3, 141.5, 136.4, 123.0, 105.9, 102.5, 101.7, 97.8, 90.7. LCMS (ESI^+^) *m/z* calcd. For [C_14_H_10_NO_5_]^+^: 294.0378, found: 294.0379.

(*Z*)-3-((4,6-Dihydroxy-3-oxobenzofuran-2(3*H*)-ylidene)methyl)quinolin-2(1*H*)-one (**13**). Compound **13** (21.8 mg, 51%) was obtained according to GPI using 2-oxo-1,2-dihydroquinoline-3-carbaldehyde and 4,6-dihydroxybenzofuran-3(2*H*)-one. Yellow solid. ^1^H NMR (400 MHz, DMSO-*d*_6_) δ 12.07 (s, 1H), 11.02 (s, 1H), 10.98 (s, 1H), 8.69 (s, 1H), 7.84 (d, *J* = 7.8 Hz, 1H), 7.54 (t, *J* = 7.7 Hz, 1H), 7.32 (d, *J* = 8.3 Hz, 1H), 7.23 (t, *J* = 7.9 Hz, 1H), 6.90 (s, 1H), 6.30 (s, 1H), 6.10 (s, 1H). ^13^C NMR (150 MHz, DMSO-*d*_6_) δ 178.6, 167.7, 167.5, 160.8, 158.5, 149.2, 140.1, 138.4, 131.3, 128.9, 124.3, 122.3, 119.4, 115.1, 102.4, 100.7, 98.0, 90.9. LCMS (ESI^+^) *m/z* calcd. For [C_18_H_12_NO_5_]^+^: 344.0535, found: 344.0540.

(*Z*)-4-Hydroxy-6-methoxy-2-((5-(pyridin-4-yl)-4,5-dihydro-1*H*-pyrazol-3-yl)methylene)benzofuran-3(2*H*)-one (**14**). Compound **14** (17.7 mg, 28%) was obtained according to GPI using 5-(pyridin-4-yl)-1*H*-pyrazole-3-carbaldehyde and 4-hydroxy-6-methoxybenzofuran-3(2*H*)-one. Yellow solid. ^1^H NMR (400 MHz, DMSO-*d*_6_) δ 13.80 (s, 1H), 11.25 (s, 1H), 8.73 (d, *J* = 5.3 Hz, 2H), 8.04 (s, 2H), 7.52 (s, 1H), 6.64 (s, 1H), 6.55 (d, *J* = 2.0 Hz, 1H), 6.17 (d, *J* = 2.0 Hz, 1H), 3.87 (s, 3H). ^13^C NMR (100 MHz, DMSO-*d*_6_) δ 178.8, 168.5, 167.7, 158.2, 147.8, 120.4, 106.4, 103.9, 97.0, 89.4, 56.2. LCMS (ESI^+^) *m/z* calcd. For [C_18_H_14_N_3_O_4_]^+^: 336.0984, found: 336.0990.

(*Z*)-2-((5-Chloro-1*H*-pyrrolo[2,3-*b*]pyridin-3-yl)methylene)-4-hydroxy-6-methoxybenzofuran-3(2*H*)-one (**15**). Compound **15** (21.2 mg, 37%) was obtained according to GPII using 6-chloro-1*H*-pyrrolo[2,3-*b*]pyridine-3-carbaldehyde and 4-hydroxy-6-methoxybenzofuran-3(2*H*)-one. Yellow solid. ^1^H NMR (600 MHz, DMSO-*d*_6_) δ 12.67 (s, 1H), 10.97 (s, 1H), 8.67 (d, *J* = 2.3 Hz, 1H), 8.32 (d, *J* = 2.2 Hz, 1H), 8.26 (d, *J* = 2.9 Hz, 1H), 7.10 (s, 1H), 6.56 (s, 1H), 6.14 (s, 1H), 3.85 (s, 3H). ^13^C NMR (100 MHz, DMSO-*d*_6_) δ 178.6, 167.7, 167.1, 157.6, 147.0, 145.9, 142.1, 132.3, 127.5, 123.8, 119.8, 107.1, 104.5, 102.4, 96.7, 89.0, 56.1. LCMS (ESI^+^) *m/z* calcd. For [C_17_H_12_ClN_2_O_4_]^+^: 343.0486, found: 343.0485.

(*Z*)-2-((2-Chloro-7*H*-pyrrolo[2,3-*d*]pyrimidin-5-yl)methylene)-4,6-dihydroxybenzofuran-3(2*H*)-one (**16**). Compound **16** (38.0 mg, 69%) was obtained according to GPII using 2-chloro-7*H*-pyrrolo[2,3-*d*]pyrimidine-5-carbaldehyde and 4,6-dihydroxybenzofuran-3(2*H*)-one. HPLC purification was applied further to use the MeOH/H_2_O eluent system. Yellow solid. ^1^H NMR (600 MHz, DMSO-*d*_6_) δ 12.91 (s, 1H), 10.85 (s, 2H), 9.44 (s, 1H), 8.21 (s, 1H), 6.99 (s, 1H), 6.28 (d, *J* = 1.7 Hz, 1H), 6.07 (d, *J* = 1.8 Hz, 1H). LCMS (ESI^+^) *m/z* calcd. For [C_15_H_9_ClN_3_O_4_]^+^: 330.0282, found: 330.0277.

### 3.5. Enzyme Inhibition Assays

The inhibitory activities of all compounds with respect to PIM1 were measured by Reaction Biology Corp. (Malvern, PA, USA) using radiometric kinase assays ([γ-^33^P]-ATP). Briefly, the enzymatic reaction mixtures contained an artificial substrate peptide, poly-Glu-Tyr (4:1), in freshly prepared base reaction buffer (20 mM hydroxyethyl piperazine ethane sulfonic acid (HEPES) of pH 7.5, 10 mM MgCl_2_, 1 mM ethylene glycol tetraacetic acid (EGTA), 0.02% Brij-35, 0.02 mg/mL bovine serum albumin (BSA), 0.1 mM Na_3_VO_4_, 2 mM dithiothreitol (DTT), and 1% DMSO). PIM1 kinase was delivered into the substrate solution and gently mixed. Each inhibitor in 100% DMSO in a serial dilution was then delivered to the reaction mixture using the acoustic dispensing system (Echo550; nanoliter range). To initiate the enzymatic reaction, ^33^P-ATP with a specific activity of 10 μCi/μL was added into the reaction mixture, which was further incubated for 2 h at room temperature. Radioactivity was then monitored by the filter-binding method after the reactions were spotted onto P81 ion exchange paper, and the filters were washed extensively in 0.75% phosphoric acid. Kinase activity data were expressed as the percent remaining kinase activity in test samples compared to vehicle (dimethyl sulfoxide) reactions. IC_50_ values and curve fits were calculated with Prism Software (GraphPad Software) at 10 different concentrations in duplicate. Staurosporine was employed as the positive control in this study.

### 3.6. Cell Proliferation Inhibition Assay

The T47D human breast cancer cell lines were purchased from Korean Cell Line Bank (KCLB, Seoul, Korea). T47D cells were then cultured in Roswell Park Memorial Institute 1640 (RPMI-1640) medium with 10% fetal bovine serum (FBS) and 1% penicillin/streptomycin, and maintained at 37 °C in a CO_2_ incubator with a controlled humidified atmosphere comprising 95% air and 5% CO_2_. All the cell culture reagents were purchased from Thermo Fisher Scientific.

Cell viability was measured with 3-(4,5-dimethylthiazol-2-yl)-2,5-diphenyltetrazolium bromide (MTT) assay using the natural product wortmannin as a positive control. At first, T47D cells were plated at a density of 3–5 × 10^3^ cells/well in 96-well plates for 24 h and then treated with either DMSO as a control or with varying concentrations of the PIM1 inhibitors (**5**, **8**, **14**, **15**, and **16**). The final concentration of DMSO in the medium was <0.1% (*v/v*). After the incubation of cells for 48 h, 20 μL of MTT solution (5 mg/mL) was added to each well for 4 h at 37 °C. The formazan crystals formed during the mixing were dissolved in DMSO (100 μL/well) by shaking for 5 min. Duplicate wells were used for each analysis although triplicate measurements would be more reliable. After 72 h, the absorbance of each solution was measured with a microplate reader at 540 nm to calculate the antiproliferative activity of the PIM1 inhibitors. Finally, the IC_50_ values were measured from the dose–response curves.

## 4. Conclusions

We discovered new potent PIM1 inhibitors of natural origin through structure-based virtual screening to find the initial hits and subsequent de novo design using the 2-methylenebenzofuran-3(2*H*)-one moiety as the molecular core. By virtue of the modified scoring function involving an accurate molecular dehydration energy term, the discovery of PIM1 inhibitors was successful to the extent that all the initial hits and the derivatives exhibited good biochemical potency ranging from the low-micromolar to nanomolar levels. In particular, the derivatives of 2-methylenebenzofuran-3(2*H*)-one with aromatic heterocycles (**14–16**) were anticipated to serve as a lead compound for the development of anticancer medicines because of the nanomolar-level inhibitory activity and the antiproliferative effect on the cancer cell lines. It was also found that the biochemical potency of the PIM1 inhibitors could be optimized by reinforcing the interactions in the ATP-binding site in such a way to overcome the increased desolvation cost for binding to PIM1. Consistent with precedent experimental findings, the formation of hydrogen bonds with the hinge region and the activation loop of PIM1 was proven to be necessary for the high inhibitory activity. Further chemical modifications are required to improve the cellular permeability and the anticancer activity.

## Data Availability

Data is contained within the article or Appendix A.

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
