# Peer review of "Structure-Based Virtual Screening and De Novo Design of PIM1 Inhibitors with Anticancer Activity from Natural Products"

_pharmaceuticals, 2021, doi:10.3390/ph14030275_

Round 1
Reviewer 1 Report
Main considerations for the synthesis of new compounds:
Line 226-231 and Scheme 1
- the authors must explain what "various aldehydes" means, nowhere are the names of the aldehydes used in the synthesis or information about their origin,
- the name "benzofuranone" is not enough, from the information provided it is not known how the authors obtained it, what was its spectroscopic characteristics and purity, it should be explained,
- it is not known which compounds were synthesized according to which of the general procedures, it should be supplemented,
- Scheme 1 should contain the general formulas of compounds A, B and C drawn as in the general formula in Table 2, and analogous designations of the substituents (R1, R2 and R3),
- in Table 2, a column may be added informing which procedure was used for the synthesis.
Line 312 - no reference that would explain why the authors selected this cell line for research, it should be supplemented.
Materials and Methods:
Line 443-446:
- which means "proper eluent"? it should be supplemented,
- why did the authors use flash column chromatography and HPLC - I have not found the obtained results anywhere in the manuscript,
Line 457-470:
- under what conditions was the TLC performed, how were the compounds purified?, it should be supplemented
Line 471 and next:
- no information on the purity of the synthesized compounds (melting point, elemental analysis or MS, HPLC), for the derivative 16 there is even no a description of 13C NMR spectrum, it should be supplemented
If the compounds are new and their synthesis has not been described so far, an additional supplement with copies of NMR spectra should be prepared, at least for the most important (most active) compounds.
Line 561- the procedure needs to be completed (the origin of the cells and the method of cultivation).
Authors should also consider the following suggestions:
- line 45-55- literature for the text should be supplemented,
- line 107-109 - the description of the structure should be expanded,
- Table 1 - please add information about the purity of the tested compounds 1-4 (pure compounds or extracts?)
- Figure 3 - please add information about the PDB ID number and a reference.
The manuscript also contains minor typographical errors, for example :
- Line 56- bisindolylmalemides should be bisindolylmaleimides,
- Line 60 – azadidole should be azaindole,
- Line 236 – EeOH should be EtOH
Line 59-60 - incorrect numbering of the literature 25 and 26.
Author Response
Line 226-231 and Scheme 1
1) The authors must explain what "various aldehydes" means, nowhere are the names of the aldehydes used in the synthesis or information about their origin.
With respect to the synthesis of PIM1 inhibitors, all the aldehydes reacted with benzofuran-3(2H)-one were purchased from commercial compound suppliers. The full names of all these aldehydes have been presented in Section 3.4 (Chemical synthesis) in the revised manuscript.
2) The name "benzofuranone" is not enough, from the information provided it is not known how the authors obtained it, what was its spectroscopic characteristics and purity, it should be explained.
Like the aldehyde reactants, all the benzofuranone derivatives used in the synthesis of PIM1 inhibitors were purchased from commercial compound suppliers. The full names of all these benzofuranone derivatives have been presented in Section 3.4 (Chemical synthesis) in the revised manuscript.
3) It is not known which compounds were synthesized according to which of the general procedures, it should be supplemented.
In accordance with the comment, we have detailed how each PIM1 inhibitor was synthesized according to General procedure I (GPI) or General procedure II (GPII) in Section 3.4 in the revised manuscript.
4) Scheme 1 should contain the general formulas of compounds A, B and C drawn as in the general formula in Table 2, and analogous designations of the substituents (R1, R2 and R3).
As requested by the reviewer, we have modified the general formula in Scheme 1 for appropriate designations of the substituents (R1, R2, and R3).
5) In Table 2, a column may be added informing which procedure was used for the synthesis.
Following the suggestion, we have added a column in Table 2 to specify the appropriate synthetic procedure (GPI and GPII) for each PIM1 inhibitor.
Line 312
6) No reference that would explain why the authors selected this cell line for research, it should be supplemented.
With respect to the antiproliferative activity of PIM1 inhibitors, T47D cell line was used in this work because it represented the breast cancer caused by PIM1. This point has been clarified on p. 9 line 321 in the revised manuscript along with an additional reference (Ref. 48).
Line 443-446:
7) Which means "proper eluent"? it should be supplemented.
To avoid the confusion, we have removed the sentence “Flash column chromatography was performed on silica gel (400-630 mesh) or CombiFlash® Rf+ system with RediSep® Rf silica columns (230-400 mesh) using a proper eluent” in the original manuscript.
8) Why did the authors use flash column chromatography and HPLC - I have not found the obtained results anywhere in the manuscript.
Actually, we used HPLC (MeOH:H2O gradient system) for purifying purpose in the synthesis of compound 16. To explain this, we have added a sentence on p. 12 line 475 in the revised manuscript.
Line 457-470:
9) Under what conditions was the TLC performed, how were the compounds purified? It should be supplemented.
With respect to the conditions of TLC, all the PIM1 inhibitors were purified by recrystallization and washing with distilled water and CH2Cl2, except for compound 16 that was purified with HPLC. This point has been clarified in Section 3.4 (Chemical synthesis) in the revised manuscript.
Line 471 and next:
10) No information on the purity of the synthesized compounds (melting point, elemental analysis or MS, HPLC), for the derivative 16 there is even no a description of 13C NMR spectrum. It should be supplemented.
As requested by the reviewer, we have provided NMR and MS data for all the synthesized compounds in Section 3.4 (Chemical synthesis) in the revised manuscript. In case of compound 16, 13C NMR data could not be prepared because it had been used up for synthesizing the other PIM1 inhibitors as well as for enzyme inhibition and cell-based assays. However, we are going to resynthesize compound 16 in the future if 13C NMR data are critically important in the evaluation of this paper.
11) If the compounds are new and their synthesis has not been described so far, an additional supplement with copies of NMR spectra should be prepared, at least for the most important (most active) compounds.
Following the suggestion, the NMR spectra for compounds 5, 10, 13, 15, and 16 have been provided in Supplementary Materials.
Line 561:
12) The procedure needs to be completed (the origin of the cells and the method of cultivation).
With respect to the cell-based assays, the T47D human breast cancer cell lines were purchased from Korean Cell Line Bank (KCLB, Seoul, Korea). T47D cells were then cultured in RPMI-1640 supplemented with 10% fetal bovine serum (FBS) and 1% penicillin/streptomycin, and maintained at 37oC in CO2 incubator with a controlled humidified atmosphere comprising 95% air and 5% CO2. To explain these, we have added a paragraph on p. 14 line 586 in the revised manuscript.
Authors should also consider the following suggestions:
13) Line 45-55- literature for the text should be supplemented.
Following the suggestion, we have cited the two references (Refs. 12 and 15) concerning the structural features of PIM1 in comparison to other kinases.
14) Line 107-109 - the description of the structure should be expanded.
When the structures of compounds 1 and 2 are compared with those of 3 and 4, the former seems to be more druggable than the latter owing to the possession of less than five hydrogen-bond donors. To place an emphasis on this point, we have added a sentence on p. 3 line 111 in the revised manuscript.
15) Table 1 - please add information about the purity of the tested compounds 1-4 (pure compounds or extracts?).
As requested by the reviewer, we have presented the purities of the four compounds in Table 1. All these compounds were purchased from InterBioScreen Ltd., the purities of which were measured with NMR spectroscopy.
16) Figure 3 - please add information about the PDB ID number and a reference.
In accordance with the comment, we have specified the PDB code of PIM1 structure used in docking simulations on p. 4 line 132 in the revised manuscript.
The manuscript also contains minor typographical errors, for example:
17) Line 56- bisindolylmalemides should be bisindolylmaleimides; Line 60 – azadidole should be azaindole; Line 236 (220) – EeOH should be EtOH.
All the typographical errors have been corrected in the revised manuscript.
Line 59-60:
18) Incorrect numbering of the literature 25 and 26.
The reference numbers of Refs. 25 and 26 have been changed accordingly in the revised manuscript.
Reviewer 2 Report
The paper by Park et al, entitled “Structure-Based Virtual Screening and De Novo Design of PIM1 Inhibitors with Anticancer Activity from Natural Products” reports interesting findings on the development of new potential anticancer agents. Even if it must be considered a preliminary study, in my judgement the paper deserves the publication on Pharmaceutics after minor revisions.
- Some of the compounds (e.g. 5, 6 and 7), reported as newly synthesized derivatives, have been already described in literature. The bibliographic references must be included in the reference list.
- Some typos are present in the manuscript:
- g. Line 60, change “azadidole” to “azaindole”; Lines 257, 259 and 260, change “chloride” to “chlorine”
- an accurate editing of English language is required.
Author Response
1) Some of the compounds (e.g. 5, 6 and 7), reported as newly synthesized derivatives, have been already described in literature. The bibliographic references must be included in the reference list.
In accordance with the comment, we have indicated that compounds 5-7 were previously reported as the inducers of NAD(P)H:quinone oxidoreductase 1 (NQO1), on p. 6 line 243 in the revised manuscript along with an additional reference (Ref. 47).
2) Some typos are present in the manuscript. Line 60, change “azadidole” to “azaindole”; Lines 257, 259 and 260, change “chloride” to “chlorine”
All the typographical errors have been corrected in the revised manuscript.
3) An accurate editing of English language is required.
Following the suggestion, the manuscript has been revised extensively to correct grammatical errors and typos.
Reviewer 3 Report
The presented article describes the possibility of the acquisition of new medicines with the potential for PIM1 inhibition. The authors were screening natural and de novo designed compounds based on molecular simulations. Then, the most active compounds were synthesized and tested against T47D cells in MTT assay. The authors estimated IC50 values of selected compounds. The measurements in this assay were only duplicate. Therefore, I recommend analyzing triplicate measurements and adding appropriate statistical analysis in Table 3.
The manuscript is logical and well written. In my opinion, it has a chance to be an interesting position for readers.
Author Response
The authors estimated IC50 values of selected compounds. The measurements in this assay were only duplicate. Therefore, I recommend analyzing triplicate measurements and adding appropriate statistical analysis in Table 3.
Following the suggestion for adding the appropriate statistical analysis, we have presented all the IC50 values as mean ± standard deviation of the two experiments in Tables 3 in the revised manuscript. Although we agree that the triplicate measurements of biochemical potency would be more desirable for the assessment of the newly found PIM1 inhibitors, the additional cell-based assays are unavailable at present because the compounds have been used up for enzyme inhibition assays and spectral analyses as well as for the previous cell-based assays. However, we are going to resynthesize the compounds in the future if the further cell-based assays are critically important in the evaluation of this paper.
Reviewer 4 Report
In the work of Park et al “Structure-Based Virtual Screening and De Novo Design of PIM1 Inhibitors with Anticancer Activity from Natural Products” the authors performed a structure-based virtual screening of products of plant origin, de novo synthesis and tested their activity as PIM1 inhibitors and against the human breast cancer cell line T47D.
Comment addressed to table 1, table 2 and table 3.
The values of IC50 represent the “mean” of values obtained in different experiments? There is no information in the table. It will be also important to present the SD or SEM.
Comment addressed to table 1 and table 2
The authors stated that staurosporine was used as positive control but the IC50 values obtained for the compound were not shown.
Comments addressed to Table 3
In material and methods section there is no information concerning a positive control. Did the authors used a positive control (e.g. Staurosporine)?
Comment concerning T47D
The authors do not explain the reasons for choosing the human breast cancer cell line T47D. That information could be interesting since there are several human breast cancer cell lines available.
Concerning 3.6. Cell proliferation inhibition assay
The description of the technique could be improved. Some information can be added:
Origin of the cell line
Culture media used and the origin of the reagents
MTT concentration used, incubation time with MTT, compound used to dissolve the formazan product.
Was a positive control used?
If the solutions of the compounds were prepared in DMSO did the authors confirmed that the maximum concentration of DMSO used was not interfering with the assay?
The Discussion could be improved and the results analyzed and compared with those of other publications.
Author Response
Comment addressed to table 1, table 2 and table 3.
1) The values of IC50 represent the “mean” of values obtained in different experiments? There is no information in the table. It will be also important to present the SD or SEM.
In accordance with the comment, all the IC50 values were summarized as mean ± standard deviation of the two experiments in Tables 1-3 in the revised manuscript.
Comment addressed to table 1 and table 2
2) The authors stated that staurosporine was used as positive control but the IC50 values obtained for the compound were not shown.
As requested by the reviewer, the IC50 value (5.6 nM) of staurosporine has been appended in Tables 1 and 2.
Comments addressed to Table 3
3) In material and methods section there is no information concerning a positive control. Did the authors used a positive control (e.g. Staurosporine)?
With respect to the measurement of the anticancer activity, the natural product wortmannin was used as a control for inhibiting the cellular proliferation in T47D cell lines. To explain this, we have added a sentence on p. 9 line 321 in the revised manuscript. The associated IC50 value of wortmannin has also been presented in Table 3.
Comment concerning T47D
4) The authors do not explain the reasons for choosing the human breast cancer cell line T47D. That information could be interesting since there are several human breast cancer cell lines available.
Among a variety of breast cancer cell lines, T47D was used in this work to measure the antiproliferative activity of the PIM1 inhibitors because it represented the breast cancer caused by PIM1. This point has been clarified on p. 9 line 321 in the revised manuscript along with an additional reference (Ref. 48).
Concerning 3.6. Cell proliferation inhibition assay
5) The description of the technique could be improved. Some information can be added: Origin of the cell line,
Culture media used and the origin of the reagents
MTT concentration used, incubation time with MTT, compound used to dissolve the formazan product.
Was a positive control used?
If the solutions of the compounds were prepared in DMSO did the authors confirmed that the maximum concentration of DMSO used was not interfering with the assay?
The T47D human breast cancer cell lines were purchased from Korean Cell Line Bank (KCLB, Seoul, Korea). T47D cells were then cultured in RPMI-1640 supplemented with 10% fetal bovine serum (FBS) and 1% penicillin/streptomycin, and maintained at 37oC in a CO2 incubator with a controlled humidified atmosphere comprising 95% air and 5% CO2. To explain these, we have added a paragraph on p. 14 line 586 in the revised manuscript.
Cell viability was measured with 3-(4,5-dimethylthiazol-2-yl)-2,5-diphenyltetrazolium bromide (MTT) assay using the natural product wortmannin as a positive control. At first, T47D cells were plated at a density of 3-5 x 103 cells/well in 96-well plates for 24 h, and then treated with either DMSO as a control or with varying concentrations of the PIM1 inhibitors (compounds 5, 8, 14, 15, and 16). The final concentration of DMSO in the medium was < 0.1% (v/v). After the incubation of cells for 48 h, 20 mL MTT solution (5 mg/mL) was added to each well for 4 h at 37oC. The formazan crystals formed during the mixing were dissolved in DMSO (100 mL/well) by shaking for 5 min. Duplicate wells were used for each analysis. After 72 h, the absorbance of each solution was measured with a microplate reader at 540 nm to calculate the antiproliferative activity of the PIM1 inhibitors. Finally, the IC50 values were measured from the dose-response curves. To present the above details of MTT assay, we have extended the second paragraph of Section 3.6 on p. 14 in the revised manuscript.
6) The Discussion could be improved and the results analyzed and compared with those of other publications.
Following the suggestion, we compared the calculated binding mode of 16 with the X-ray crystal structures of PIM1 in complex with the known inhibitors. It was found that the inhibitory action of 16 was similar to those of CX-4945 and Ro-3306 in the involvement of hydrogen-bond interaction with Asp186. To explain this, we have added a sentence on p. 8 line 300 in the revised manuscript.
Round 2
Reviewer 1 Report
In the materials and methods section, the authors should add information about the method and apparatus used to perform the LCMS analysis (ESI +).
“In case of compound 16, 13C NMR data could not be prepared because it had been used up for synthesizing the other PIM1 inhibitors as well as for enzyme inhibition and cell-based assays. However, we are going to resynthesize compound 16 in the future if 13C NMR data are critically important in the evaluation of this paper.”
Such a comment is not sufficient. When testing new substances, the first and most important step is a complete and clear analysis of the substance. In order to use it for further syntheses or biological research, the researcher must be sure what compound he is using. The authors did not add information about the melting point of the tested compounds and the result of the LS MS analysis is not sufficient to assess the purity. A HR MS result given with adequate accuracy would be more reliable.
Author Response
1) In the materials and methods section, the authors should add information about the method and apparatus used to perform the LCMS analysis (ESI+).
With respect to the method and apparatus, high-resolution mass spectra were obtained by the ESI method from KAIST Research Analysis Center (Daejeon, Korea), which involved the mass measurement with Quadrupole-TOF MS system. To explain this, we have added a sentence on p. 12 line 460 in the revised manuscript.
“In case of compound 16, 13C NMR data could not be prepared because it had been used up for synthesizing the other PIM1 inhibitors as well as for enzyme inhibition and cell-based assays. However, we are going to resynthesize compound 16 in the future if 13C NMR data are critically important in the evaluation of this paper.” Such a comment is not sufficient. When testing new substances, the first and most important step is a complete and clear analysis of the substance. In order to use it for further syntheses or biological research, the researcher must be sure what compound he is using. The authors did not add information about the melting point of the tested compounds and the result of the LS MS analysis is not sufficient to assess the purity. A HR MS result given with adequate accuracy would be more reliable.
In accordance with the comment, we have presented the HRMS results for all the synthesized compounds in Section 3.4 (Chemical synthesis) in the revised manuscript.
Reviewer 3 Report
I hope, you will provide triplicate analyses of IC50 in the close future.
Author Response
In accordance with the comment, we have indicated that triplicate measurements would be more reliable in cell-based assays, on p. 15 line 603 in the revised manuscript.
Reviewer 4 Report
According to the reviewers' comments, the authors altered and improved the manuscript.
However, the authors could still improve the manuscript.
The origin of the reagents and culture medium used could be introduced into the material and methods section.
The authors could standardize the values in the tables with regard to the number of decimal places of the numbers presented.
Author Response
1) According to the reviewers' comments, the authors altered and improved the manuscript. However, the authors could still improve the manuscript. The origin of the reagents and culture medium used could be introduced into the material and methods section.
With respect to the cell-based assays, T47D cells were cultured in Roswell Park Memorial Institute 1640 (RPMI-1640) medium with 10% fetal bovine serum (FBS) and 1% penicillin/streptomycin. This has been clarified on p. 14 line 590 in the revised manuscript. We have also indicated that all the cell culture reagents were purchased from Thermo Fisher Scientific, on p. 14 line 593.
2) The authors could standardize the values in the tables with regard to the number of decimal places of the numbers presented.
Following the suggestion, we have expressed all the SD values in Tables 1-3 with a single significant figure.